# Efficient intersystem crossing and tunable ultralong organic room-temperature phosphorescence via doping polyvinylpyrrolidone with polyaromatic hydrocarbons

Guangxin Yang[1], Subin Hao[1], Xin Deng[1], Xinluo Song[1], Bo Sun [2] ✉, Woo Jin Hyun [3], Ming-De Li [1,4] ✉ & Li Dang [1] ✉

Polymer-based pure organic room-temperature phosphorescent materials have tremendous advantages in applications owing to their low cost, vast resources, and easy processability. However, designing polymer-based room-temperature phosphorescent materials with large Stokes shifts as key requirements in biocompatibility and environmental-friendly performance is still challenging. By generating charge transfer states as the gangplank from singlet excited states to triplet states in doped organic molecules, we find a host molecule (pyrrolidone) that affords charge transfer with doped guest molecules, and excellent polymer-based organic room-temperature phosphorescent materials can be easily fabricated when polymerizing the host molecule. By adding polyaromatic hydrocarbon molecules as electron-donor in polyvinylpyrrolidone, efficient intersystem crossing and tunable phosphorescent from green to near-infrared can be achieved, with maximum phosphorescence wavelength and lifetime up to 757 nm and 3850 ms, respectively. These doped polyvinylpyrrolidone materials have good photo-activation properties, recyclability, advanced data encryption, and anti-counterfeiting. This reported design strategy paves the way for the design of polyvinylpyrrolidone-based room-temperature phosphorescent materials.

Luminescent materials are of significance for both commercial and scientific purposes[1–15]. Among those, ultra-long organic room-temperature phosphorescent (ORTP) materials have attracted considerable attention in information storage and encryption[16,17], advanced anticounterfeiting[18,19], sensing[20,21], optoelectronics[22,23] and bioimaging and diagnostis[24–28] due to luminescent time. According to the Jablonski diagram, there are two requirements to achieve good phosphorescence: one is to promote the intersystem crossing (ISC)

[1]College of Chemistry and Chemical Engineering, Key (Guangdong-Hong Kong Joint) Laboratory for Preparation and Application of Ordered Structural Materials of Guangdong Province, Shantou University, Guangdong 515063, P. R. China. [2]State & Local Joint Engineering Research Center for Ecological Treatment Technology of Urban Water Pollution, College of Life and Environmental Science, Institute for Eco-environmental Research of Sanyang Wetland, Wenzhou University, Wenzhou, Zhejiang 325035, P. R. China. [3]Department of Materials Science and Engineering, Guangdong Technion-Israel Institute of Technology, Shantou, Guangdong 515063, China. [4]Chemistry and Chemical Engineering Guangdong Laboratory, Shantou 515031, China. ✉e-mail: sunbo@wzu.edu.cn; mdli@stu.edu.cn; ldang@stu.edu.cn

rate from excited singlet state to excited triplet state; the other is to enhance the stabilities of triplet exciton of organic molecules. To this end, strategies including crystallization engineering[29,30], H-aggregation[31], metal-organic frameworks[32], host-guest systems[33–35], integration with halogen bonding[36,37] and doping in polymer[38–40] have been proposed to obtain ORTP. However, most of the ORTP materials reported to date, exhibit only green and yellow phosphorescent emission, in contrast, white or red long-persistent emission is rarely reported[1,2]. Moreover, the phosphorescence lifetimes of the reported red ORTP materials are very short, probably resulting from the energy gap law, limiting the practical applications. To establish ORTP materials with long-wavelength emission necessarily comes from materials with low energy level triplet state ($T_1$). Lower $T_1$ brings two major issues to phosphorescence. One is that lower $T_1$ implies a larger band gap ($\Delta E_{st}$) between $S_1$ and $T_1$, which is not favorable to intersystem crossing (ISC) of excitons, making it difficult to observe phosphorescence. The other is that lower $T_1$ is more likely to lead to non-radiative depletion of excitons, which results in a significant reduction in the lifetime and intensity of the phosphorescence.

Fortunately, an energy transfer passage could be unlocked by generating intermolecular charge transfer states as an energy bridge, and usually, $T_1$ could be stabilized in doped systems. In our previous study, the efficient red long-persistent emission from purely organic doped molecules was obtained[41]. These materials can be easily fabricated by mixing pyrene (electron donor) and benzophenone (electron acceptor) using the melt-casting method. The mechanism study shows that it is the charge-transfer (CT) states that facilitate the ISC of the guest, resulting in room-temperature phosphorescence (RTP) from guest triplet states (Fig. 1a). This study gives us the hint that to make good use of polyaromatic hydrocarbons as the guests with very low $T_1$ energy levels due to the highly conjugated nature of these molecules, which is crucial for long-wavelength phosphorescence. But at the same time, they also have very high $S_1$ energy levels, which will result in a very large $\Delta E_{st}$, and ISC is difficult to occur. The above problems can be easily solved by constructing the host-guest charge transfer states, thus ORTP materials with long-wavelength emission can be prepared. Practically,

the poor reproducibility and flexibility of crystal-based ORTP materials limits their applications. It is well known that polymer-based afterglow materials show good flexibility and transparency[6,42]. Moreover, polymer-based ORTP materials are suitable for large-scale production through solution processing which is beneficial for commercialization[19,22]. To avoid the cumbersome preparation process of modifying polymers with electron donor and acceptor groups to achieve good luminescence[43], it is necessary to apply a doped host-guest strategy in the field of polymer-based ORTP materials by constructing materials with aromatic hydrocarbons as the guests (electron donor) and appropriate polymer as the host (electron acceptor) (Fig. 1b).

Polymer-based pure ORTP materials have become a rapidly growing research field, and various preparation strategies have been developed, such as doping[44–46], copolymerization[47–49], homopolymerization[50–52], and host–guest inclusion[53–55]. However, the overall performance of RTP, including phosphorescence lifetime and quantum efficiency, still needs to be further optimized. More to the point, only sporadic examples have emitted ultra-long wavelength afterglows so far[56–58], and most of them are organic crystals[1,2,59–61]. It is still very challenging to achieve polymer-based pure ORTP materials with long wavelength emission.

In this work, we constructed the tunable luminescent polymer-based ORTP materials by using highly conjugated polyaromatic hydrocarbons (PAHs) molecules as guests and polyvinylpyrrolidone (PVP) as the host. Materials with a series of phosphorescence wavelengths from 550 nm to 757 nm and lifetimes from 139 ms to 3850 ms can be prepared (Fig. 1c). In addition, these polymer-based ORTP materials show excellent photoactivation properties and thus can be used as programmable labels to achieve dual fluorescent-phosphorescent advanced anti-counterfeiting functions.

## Results

First of all, when different fluorescent PAHs molecules as guests (Fig. 2a) and PVP (Mw = 40,000-45,000 g/mol) as host are mixed at a mass ratio of 1% to prepare doped films, and fluorescence emission under 365 nm UV excitation is observed (Fig. 2b UV on and

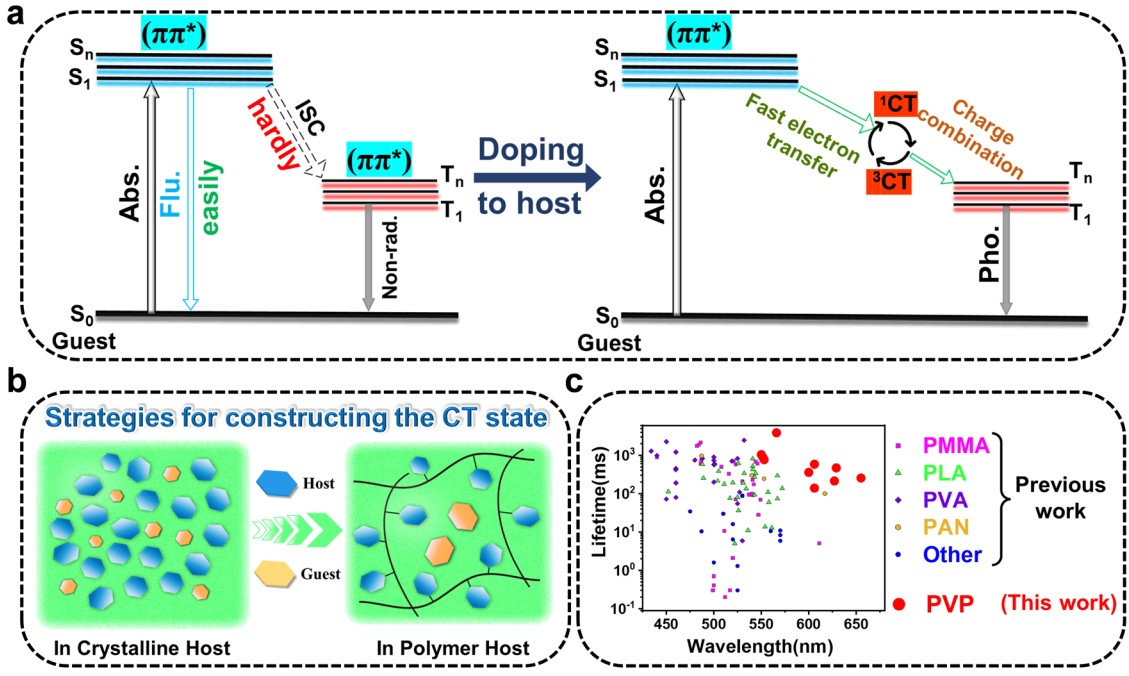

**Fig. 1 | Rational design of polymer-based ORTP material with long wavelength emission. a** The ORTP mechanism of the host-guest system in organic crystals was proposed by our group in a previous work. **b** Strategy used in this work for constructing polymer-based ultralong ORTP materials. **c** Phosphorescence performance distribution of polymer-based ORTP materials. Abs. is Absorption, Flu. Is Fluorescence, Non-rad. is Non-irradiation, Pho. is Phosphorescence.

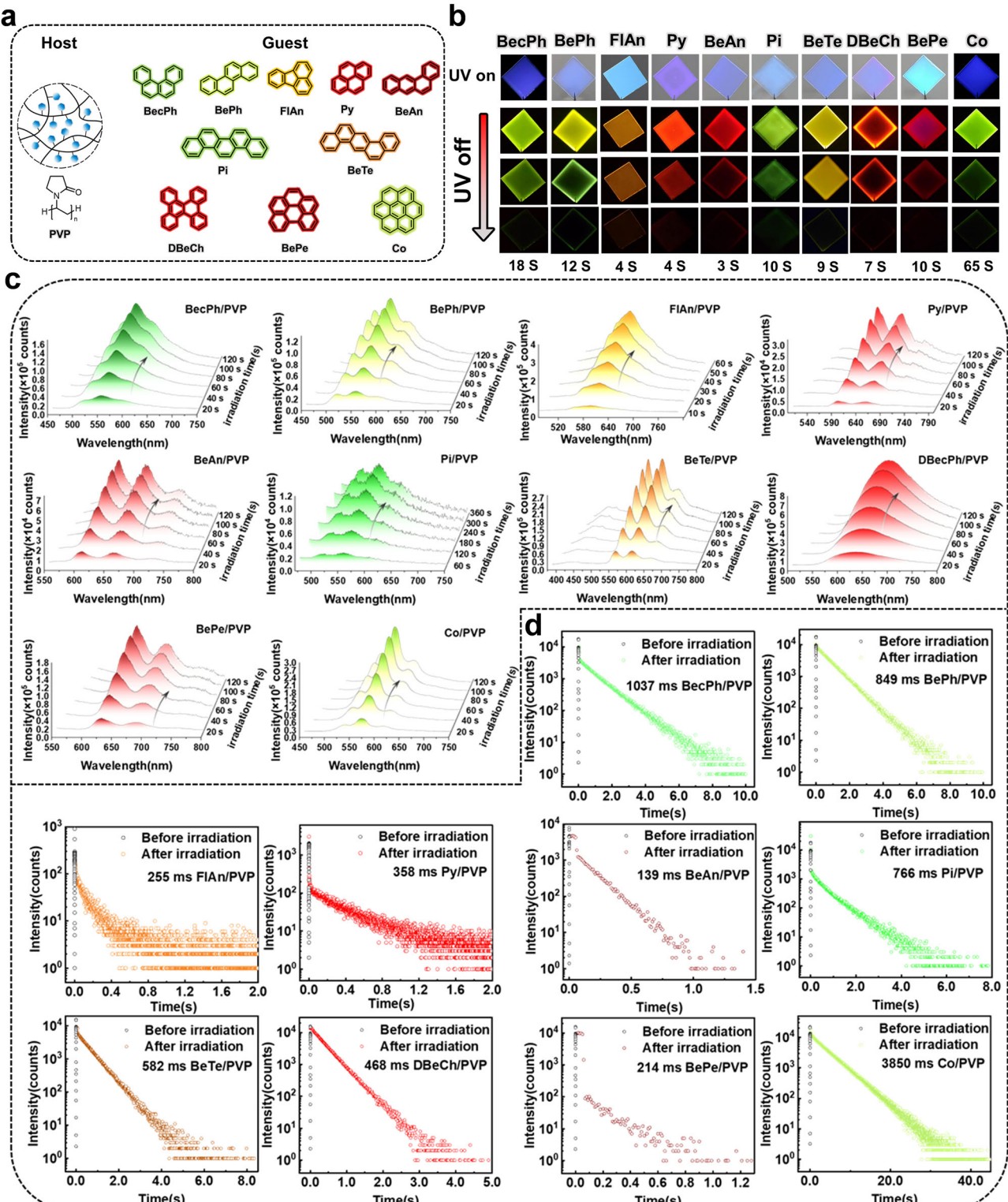

**Fig. 2 | Photo-induced RTP properties of the PVP films doped with different guest molecules. a** Molecular structures of the host and guest molecules. **b** Fluorescence and phosphorescence images of PVP films doped with the different guest molecules under UV light at 365 nm, with the name of the guests at the top of the image and the duration of the afterglow observable to the naked eye at the bottom of the image. **c** Delayed emission spectra of these doped PVP films after photoactivation under UV irradiation at 365 nm at different times, delay time: 1 ms. **d** Emission decay curves of the doped PVP films before and after photoactivation.

Supplementary Fig. 1). Compared to a blank PVP film with irradiation by 365 nm UV for a short time (Supplementary Fig. 2), intense different color afterglows from green to NIR are manifested after irradiation for a longer time (Fig. 2b UV off and Supplementary movie 1-10). As can be seen in Fig. 2c, the RTP of these polymer-based materials is gradually enhanced when increasing UV irradiation time and their maximum emission peaks are listed in Table 1. Moreover, the luminescent lifetimes gradually increase with longer irradiation time (Fig. 2d). In the meantime, good to excellent phosphorescence quantum yields ($\Phi_p$) of these activated films are determined and listed in Table 1.

**Table 1 | Photophysical data of the polymer-based ORTP materials**

| Sample | Fluo. | | | Phos. | | |
|---|---|---|---|---|---|---|
| | $\lambda_{em}$(nm) | $\varphi_F$(%) | $\tau$(ns) | $\lambda_{em}$(nm) | $\varphi_P$(%) | $\tau$(ms) |
| BecPh/PVP | 422 | 14.3 | 40 | 550 | 27.5 | 1037 |
| BePh/PVP | 405 | 13.9 | 15 | 552 | 12.9 | 849 |
| FlAn/PVP | 467 | 22.2 | 41 | 585 | 7.7 | 255 |
| Py/PVP | 416 | 11.2 | 62 | 600 | 8.4 | 358 |
| BeAn/PVP | 413 | 7.1 | 36 | 606 | 6.2 | 139 |
| Pi/PVP | 425 | 7.6 | 28 | 553 | 10.3 | 766 |
| BeTe/PVP | 421 | 15.4 | 28 | 606 | 13.8 | 582 |
| DBeCh/PVP | 418 | 13.4 | 12 | 629 | 24.3 | 468 |
| BePe/PVP | 432 | 19.5 | 67 | 627 | 7.3 | 214 |
| Co/PVP | 447 | 11.1 | 86 | 566 | 10.1 | 3850 |

Ex. of Fluo.: 375 nm; Ex. of Phos.: 365 nm; Delayed time: 1 ms.

In addition, the delayed emission spectra of PAHs doped PVP films (Fig. 2c) are consistent with the delayed emission spectra of PAHs in a dilute solution of 2-methyltetrahydrofuran at 77 K (Supplementary Fig. 3). These results suggest that the doped materials emit light from the triple excited state of the guest molecules which are dispersed in the polymer (PVP). Although the PAHs are potential phosphorescent luminogens, PVP the host is necessary for the efficient phosphorescence of PAHs. Generally, the host material can act as a sealer to prevent the triplet excitons from being quenched by oxygen[48]. Therefore, we investigated the influence of $O_2$ on the luminescent behavior of BecPh/PVP and BePh/PVP films and found that the RTP was further enhanced under vacuum conditions (Supplementary Fig. 4). The quenching of the phosphorescence of the pristine samples without photoactivation may be caused by the energy transfer from the triplet excitons of the guest compound to the $O_2$ molecule in the PVP. These results also indicate that the luminescence involves triplet states and explain the results in Figs. 2c and 2d that the luminescent intensities and lifetimes are enhanced along with the increase of irradiation time allowing $O_2$ consumption. In addition, four annealing temperatures, 20 °C, 40 °C, 60 °C and 80 °C were selected for thermal analysis of the doped films. As the annealing temperature increases, the required annealing time decreases, which is in accordance with the time-temperature equivalence principle. In order to achieve faster annealing results, an annealing temperature of 80 °C and an annealing time of 5 min were chosen. These results fully demonstrate that these doped films possess satisfactory photoactivated RTP performance.

In order to clarify what promotes the luminescent switch from fluorescence to phosphorescence for doped PAHs, ultrafast femtosecond transient absorption (fs-TA) spectra are employed to investigate the distinct luminescence behaviors of the photo-activated polymer film doped with Py and BePe (Fig. 3). When using 365 nm as a pump beam, the guest is excited, and the excitation of PVP can be mostly avoided (Supplementary Fig. 5). Figure 3a shows two obvious excited state absorption (ESA) peaks for Py dispersed in PMMA. The band at 470 nm is ascribed to the absorption from the excited singlet state and the peak at 510 nm is assigned to the absorption of Py's excimer. The fs-TA spectra of Py dispersed in PMMA are very close to that in MeCN solution (Supplementary Fig. 6). A dramatic difference can be seen for Py-doped PVP film. For the CT between host and guest in Py/PVP doped films, the population and transformation of the CT absorption is usually very fast. As shown in (Supplementary Fig. 7), the transient absorption peaks gradually rise up and have a red-shift from 433 to 441 nm for Py/PVP film and from 521 to 525 nm for BePe/PVP film at the beginning delay times. These steps are attributed to the CT process in the doped film. Generally speaking, the host is the vast majority, and the guest content is very small in the doped systems. Therefore, it is

usually very difficult to observe the cationic or anionic radicals by the transient absorption when it undergoes intermolecular charge transfer in the excited state. But in our system, the conjugation of guest molecules is very good and aromatic, thus the cationic radical (441 nm) of the guest is directly observed by fs-TA. However, the host molecule is not conjugated, thus the molar extinction coefficient of its anionic radicals will be relatively small, and it is difficult to detect the corresponding anionic cations. Nevertheless, the detection of the cationic radical signal of the guest molecule in the transient absorption spectra can further prove that the intermolecular charge transfer process in the excited state has occurred between host and guest in the doped systems. After 72.2 ps, an ESA peak at 441 nm obviously shows up (Fig. 3b and Supplementary Fig. 8). Based on our previous study, this 441 nm peak is assigned to the absorption of the Py radical cation[41]. Subsequently, the long-lived triplet state with the main ESA peak around 370 nm is generated along with the decay of 441 nm and 470 nm absorption peaks[62,63]. Finally, the transient species decays gradually to zero. From the kinetics fitting, the 370 nm peak for Py-doped PVP film decays much slower than that for Py dispersed in PMMA (Supplementary Fig. 9), which is consistent with the phenomenon that Py-doped PVP film shows RTP feature. More convincingly, similar results are observed in BePe/PVP films. For BePe dispersed in PMMA (Fig. 3c), the peak at 422 nm is ascribed to the absorption from the excited singlet state of BePe, and the band at 514 nm is assigned to the absorption of BePe's excimer. The fs-TA spectra of BePe dispersed in PMMA are very close to the spectral evolution of BePe in the dichloromethane solvent (Supplementary Fig. 10). When BePe is dispersed into PVP, the fs-TA spectra show a new ESA peak at 525 nm (Fig. 3d), which is attributed to the absorption of the BePe radical cation. This can also prove the intermolecular charge transfer from BePe molecules to side groups of PVP. In addition, BecPh/PVP and DBeCh/PVP films are also explored by fs-TA spectra (Supplementary Fig. 11), which are similar to those of Py/PVP and BePe/PVP. All these results show that it is charge transfer states facilitating the ISC processes of the PAHs/PVP.

To further confirm the intermolecular CT process between the PAHs and PVP polymer, theoretical calculations are carried out based on density functional theory (DFT) at the B3LYP/6-31 G(d) level. For this doping system, the host is a polymer. Since it is very difficult to calculate and analyze the molecular orbital energy level of polymer, we select a monomer (1-vinyl-2-pyrrolidinone (VP)) of PVP as the template for analysis. The results show that VP has higher highest occupied molecular orbital (HOMO) and lower lowest unoccupied molecular orbital (LUMO) energy levels than PVP. As the unit number of VP increases, the HOMO of the VP combinations gradually increases while the LUMO gradually decreases (Fig. 4a). The energy differences between the HOMOs of the guests and LUMO of PVP could be small enough for intermolecular charge transfer[64,65] (Supplementary Fig. 12). Since the LUMOs of the guests have higher energy than the LUMO of PVP, charge transfer can occur from the excited guests to PVP after 365 nm UV lighting. To validate the above experimental data, the excited state calculations have been performed on the guests and the pair. The adiabatic energy maps can be drawn based on the optimized geometric structure of the singlet and triplet excited states, and the spin-orbit coupling (SOC) constants are calculated based on the optimized $S_1$.

The calculated results show that the $\Delta E_{S1\rightarrow T1}$ values of the guests are large, and the SOC values are quite small (Table 2), showing that ISC is difficult to occur. This result is consistent with the spectroscopic results that the guest molecules are fluorescent molecules rather than phosphorescent ones. But when the fs-TA spectra of such molecules were tested, anomalies were found. Taking the fs-TA spectra of Py and BePe as an example, the characteristic absorption peak of the excited triplet state of Py at 411 nm in MeCN and the excited triplet state of BePe at 470 nm for BePe in DCM can be detected. In addition, similar results were obtained for other PAHs molecules when fs-TA

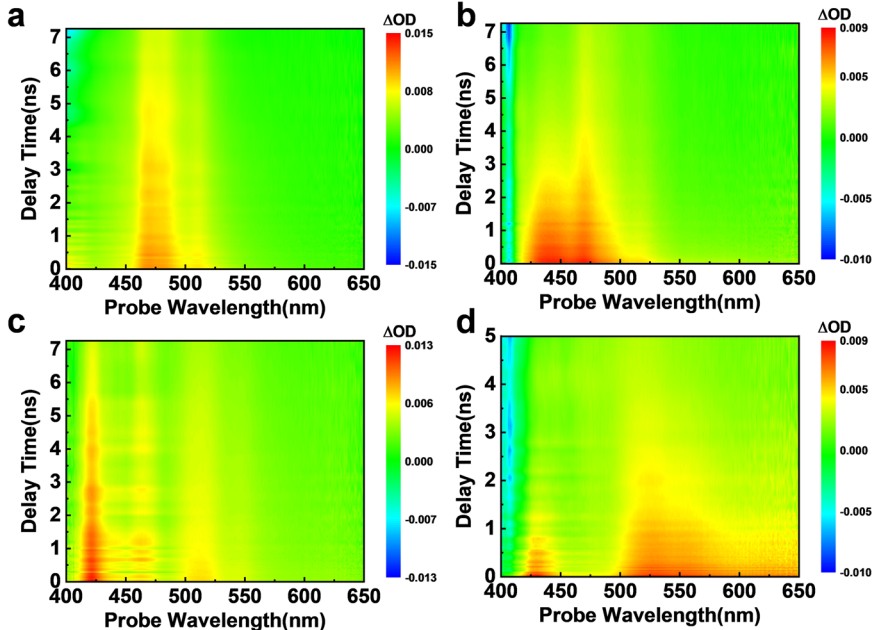

**Fig. 3 | Validation of charge transfer between polymer PVP and guest molecules.** Blank controls for the mapping image of fs-TA spectra measured from Py/PMMA (**a**) and BePe/PMMA (**c**) film, the mapping image of fs-TA spectra of Py/PVP (**b**) and BePe/PVP (**d**) films after activation with 365 nm UV light, respectively.

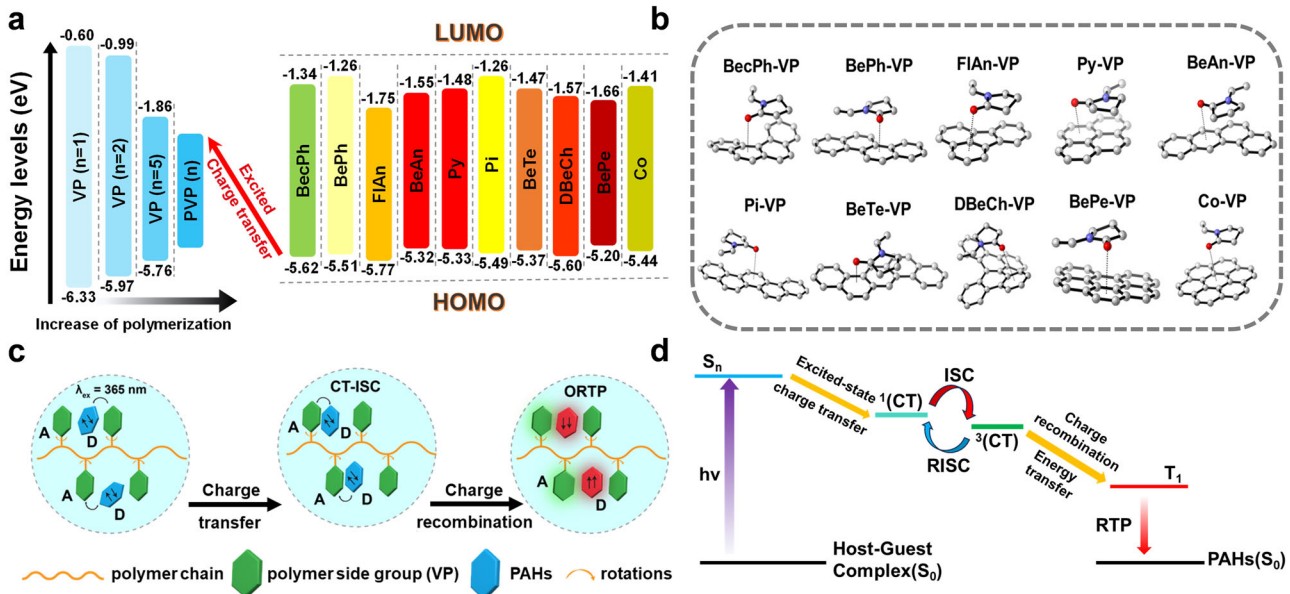

**Fig. 4 | Mechanistic investigations of polymer-based ORTP materials under ambient conditions. a** Energy level diagram of different PAHs and PVP. **b** Optimized ground-state molecular configurations for the PAHs/VP pair. **c** Schematic illustration of the CT state between PAHs and polymer side groups assisting PAHs to undergo the ISC process. **d** Proposed mechanism of ORTP of this host-guest system.

experiments were performed. This implies that PAHs molecules can actually undergo ISC to produce the corresponding excited triplet states, which is completely inconsistent with the calculated results. Through literature review[66–70] and further experiments, we have revealed the reason why PAHs can undergo ISC. For PAHs molecules, whose structures are all planar, they are very prone to generate the excimers upon excitation, which allows them to undergo ISC, leading to the generation of the corresponding triplet excitons. To test the hypothesis, we have measured the fs-TA spectra of Py with different concentrations in MeCN. The results found that when the concentration of Py is very small, the transient absorption signal of excited triplet state excitons of Py is very small because it is very difficult to form the excimers. However, with the increasing concentration of Py, it is more likely to produce excimers, so the intensity of the characteristic absorption peak of Py's excimer (508 nm) obviously increases, and the characteristic absorption peak of excited triplet states (411 nm) of Py is also obviously enhanced (Supplementary Fig. 13a). In addition, we also tested the fs-TA spectra of BePe with different concentrations in DCM, and the same results can be obtained (Supplementary Fig. 13b). But even so, the formation of the excimers based on the conjugated molecule promotes the ISC process only to a small extent, which explains the fact that most of these guests show very weak RTP when

doped with polymers such as PVA, PMMA, etc. For example, the doped film of Py/PMMA produces a very weak red phosphorescence. However, when excited state intermolecular charge transfer is formed between the host and PAHs molecules, the ISC process of PAHs molecules is greatly facilitated, resulting in the generation of a large number of triplet excitons. This can be verified not only from the small-molecule doping systems that we have previously studied[41]. But more importantly, the triplet state yield of the guest PAHs in the polymer system can also be dramatically boosted by its intermolecular charge transfer with the side groups of the PVP. This makes the RTP of Py/PVP doped films much stronger than that of Py/PMMA doped films at the same doping ratio (Supplementary Fig. 14). The calculated $\Delta E_{st}$ and SOC values of the complex satisfy the optimal conditions for the ISC (Fig. 4b and Supplementary Table 1). Therefore, the charge transfer state opens a new trajectory for the ISC of the guests, which will dra-

matically improve the ISC process of PAHs guest molecules. More importantly, the calculated triplet state energies essentially match the phosphorescence colors, which also confirms that the phosphorescence emission originates from the guest triplet state.

Based on the luminescent phenomena and theoretical studies, the RTP mechanism of these doped films is becoming clear. As shown in Fig. 4c, when the polymer side groups and guest molecules form a unit, the charge transfer from guests to PVP side groups is facilitated. When guests are excited, a charge transfer state is formed in a short time. This host-stabilized charge transfer state can transform into a triplet charge transfer state, and then give a triplet state for guests through charge recombination. This is comparable to facilitating the ISC process of the guest molecules, resulting in a substantial increase in the number of guest triplet state excitons. Eventually, as the photo-activation time is prolonged, the $O_2$ in the PVP is completely consumed, and RTP occurs after removing the excitation light (Fig. 4d).

It is well known that PVP has shown good prospects in the fields of medicine, food, and cosmetics due to its excellent solubility and biocompatibility. After doping with PAHs, it still exhibits excellent performance. If PVP solutions with different guests are applied as coatings on transparent plastic paper, it is difficult for the naked eye to see the written pattern clearly after the solvent evaporates. However, under UV irradiation, the patterns written on the material show a corresponding strong fluorescence. After 2 min of irradiation, the UV light is removed, and the patterns are drawn with corresponding afterglows. (Fig. 5a and Supplementary Fig. 15). In addition, this material can be used for the manufacturing of 3D anti-counterfeiting objects. These 3D objects appear transparent in sunlight and exhibit the same luminescent properties as before after removing excitation (Fig. 5b).

Moreover, the activation-deactivation process is repeatable because there is no significant change in the RTP spectrum, luminescent intensity, and lifetime (Supplementary Fig. 16). Inspired by the switchable RTP and steady-state luminescence characteristics, the Py/PVP and FlAn/PVP are exploited as an erasable transparent film for the

**Table 2 | Energies of $S_1$, $T_1$, $T_2$, energy gaps between $S_1$ and $T_1/T_2$ states (in eV) and SOC (cm$^{-1}$) of PAHs**

|  | $S_1$ | $T_1$ | $T_2$ | $\Delta E_{(S_1 \to T_1)}$ | $\Delta E_{(S_1 \to T_2)}$ | $SOC_{(S_1 \to T_1)}$ | $SOC_{(S_1 \to T_2)}$ |
|---|---|---|---|---|---|---|---|
| BecPh | 3.71 | 2.52 | 3.11 | 1.19 | 0.60 | 0.03 | 0.00 |
| BePh | 3.79 | 2.49 | 3.24 | 1.30 | 0.55 | 0.03 | 0.00 |
| FlAn | 3.39 | 2.37 | 2.63 | 1.02 | 0.76 | 0.01 | 0.00 |
| Py | 3.72 | 2.12 | 3.46 | 1.60 | 0.26 | 0.00 | 0.04 |
| BeAn | 3.40 | 2.05 | 3.04 | 1.35 | 0.36 | 0.01 | 0.04 |
| Pi | 3.60 | 2.48 | 2.95 | 1.12 | 0.65 | 0.04 | 0.00 |
| BeTe | 3.43 | 2.24 | 3.09 | 1.19 | 0.34 | 0.03 | 0.00 |
| DBeCh | 3.58 | 2.41 | 3.12 | 1.17 | 0.46 | 0.02 | 0.00 |
| BePe | 3.28 | 2.03 | 3.03 | 1.25 | 0.25 | 0.00 | 0.05 |
| Co | 3.23 | 2.33 | 2.97 | 0.90 | 0.26 | 0.04 | 0.00 |

B3LYP/6-31 G* and TD-B3LYP/6-31 G*.

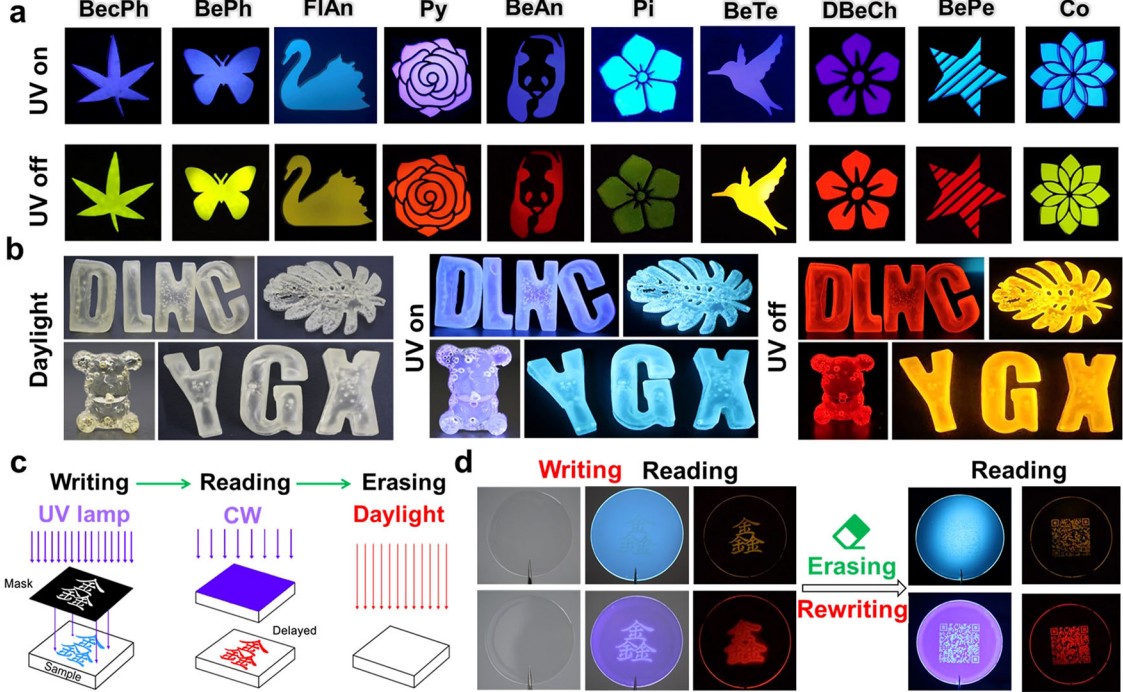

**Fig. 5 | Demonstration of polymer-based ORTP materials for data encryption, 3D printing, and programmable labels under environmental conditions.** **a** Luminescence images of various patterns made from the polymer-based ORTP materials as coatings under UV lamp (365 nm) excitation and after removing the UV light (the names of the relevant guests are labeled at the top of the images). **b** Luminescent images of 3D casting objects of Py/PVP and FlAn/PVP under daylight, on UV lamp (365 nm) excitation, and after removal of UV light. **c** Schematic of writing, reading, and erasing procedure of programmable label. **d** Luminescent images of programmable labels based on Py/PVP and FlAn/PVP film.

Programmable Label. Figure 5c shows the overview of the writing, reading, and erasing procedure of the programmable label. By mask illumination of the samples, any pattern can be applied as phosphorescent information storage. Py/PVP and FlAn/PVP films are covered with a mask A of the Chinese character '鑫' and then irradiated by the 365 nm UV light (9 mWcm⁻²) for 120 seconds. Remove the mask after the writing is completed, and the corresponding afterglow of the Chinese character '鑫' appears on the doped film when the information is read. It is worth noting that this pattern is invisible to the naked eye under ambient conditions. Under the excitation of 365 nm UV light, the white Chinese character '鑫' can be observed on both FlAn/PVP and Py/PVP films. Moreover, the resulting picture could be retained for several seconds due to the RTP properties of FlAn/PVP and Py/PVP. When ceasing the excitation light source, FlAn/PVP and Py/PVP display yellow and red written patterns, respectively, and the background turns black. Afterward, the pattern is erased via annealing at 80 °C for 5 min, and a renewed film is obtained by replacing mask A with mask B of quick response code and repeating the above procedures, the quick response code can be clearly printed. Interestingly, using the mobile phone to scan the patterns highlighted by the yellow and red RTP can facilely access the webpage information (Fig. 5d and Supplementary Fig. 17). In order to further enhance the application prospects of these materials, Py/PVP and FlAn/PVP solutions are directly coated on paper. After the solvent has evaporated, the previous operation is repeated. When the excitation light source is removed, clear writing patterns can still be obtained (Supplementary Fig. 18).

## Discussion

In summary, we have innovatively developed a strategy to assist PAHs that lack any significant means of SOC for the ISC process to achieve efficient ORTP by using the construction of charge transfer states with PVP polymer. Specifically, ORTP from green to near-infrared is easily achieved by doping different PAHs in PVP. Among them, the RTP emissions of Py/PVP, BeAn/PVP, DBeCh/PVP, and BePe/PVP materials are all in the red region with RTP lifetimes of 358, 139, 468 and 214 ms, and afterglow durations of 4, 3, 7 and 4 s, respectively, which are very promising red RTP emitters with exceptional performance. The good RTP property is attributed, from spectroscopy and computational studies, to intermolecular charge transfer (from PAHs to side groups of PVP), which opens a new gangplank to promote the ISC process of PAHs that leads to the emission from excited triplet states of PAHs. More importantly, these high-performance ORTP materials with different colors show great potential in applications from information storage and data encryption to programmable labels and bioimaging. Our study further complements the insufficiencies in the interpretation of the ORTP mechanism in the polymer-based doping system and serves as a guideline for the future design of large Stokes shift ORTP materials with long wavelength and lifetime emission.

## Methods

### Material science

Polyvinylpyrrolidone (PVP), Benzo[c]phenanthrene (BecPh, 95%), Benzo[a]phenanthrene (BePh, 99%), Pyrene (Py, 98%), Fluoranthene (FlAn, 98%), Benz[a]anthracene (BeAn, 98%), Picene (Pi, 99%), Dibenz[a,h]anthracene (BeTe, 98%), Dibenzo[g,p]chrysene (DBeCh, 98%), Benzo[ghi]perylene (BePe, 98%), and Coronene (Co, >94%) were obtained from J&K Scientific Ltd. All other solvents and reagents were purchased with analytical grade and used without further purification.

### Preparation of the polymer-based RTP film

Precursor solution synthesized by mixing different polyaromatic hydrocarbons (PAHs) as the guests and the PVP as host according to the mass ratio of 1:100 in organic solution. The prepared solution was added dropwise on the quartz sheet, and the corresponding polymer-based RTP films could be obtained after all the solvent evaporated.

### UV-vis absorption experiments

The UV-Vis absorption spectra of samples were recorded by a spectrometer (PE Lambda950) from PERKINELMER company.

### Photoluminescence (PL) experiments

The PL spectra were recorded by a spectrometer (FLS980) from Edinburgh Instruments. Doped films are used for all tests, and the amounts of samples for PL measurement are similar. Time-resolved PL decay kinetics and PL spectra were recorded in room temperature by the FLS980 from Edinburgh company.

### Femtosecond transient absorption (fs-TA) experiments

The fs-TA experiments were carried out by employing a femtosecond regenerative amplified Ti: sapphire laser system in which the amplifier was seeded with the 84 fs laser pulses from an oscillator laser system. The probe pulse was generated by ~5% of the amplified 800 nm laser pulses to produce a white-light continuum (350–800 nm) in a $CaF_2$ crystal. After that this probe laser was split into two parts before pathing the sample. One probe laser went through the sample while the other probe laser went to the reference spectrometer in order to monitor the fluctuations of the probe laser intensity. The laser 365 nm with 0.2 mW power was used as the excitation wavelength. The single-wavelength dynamics fitting was performed based on equation (1). The $\tau$ is referred to as delay time, and $\tau_0$ is zero time. The $\tau_p$ and A are the instrument response value and the proportion of species, respectively.

$$S(t) = e^{-\left(\frac{t-t_0}{t_p}\right)^2} * \sum_i A_i e^{-\frac{t-t_0}{t_i}} \tag{1}$$

### Theoretical calculations

The theoretical calculations are carried out using the Gaussian 16 software packages. B3LYP is selected as the function, and 6-31 g(d) is selected as the bases set to optimize the structures and analyze the energy level, the cartesian coordinates for the complexes (VP, (VP)₂, (VP)₅, (VP)₁₀, BecPh, BePh, FlAn, Py, BeAn, Pi, BeTe, DBeCh, BePe, Co) calculated in this study were put in the Supplementary Data 1. The structures of the host-guest complexes are shown in Multiwfn software.

## Data availability

The authors declare that the data supporting the results of this study are available within the article and its Supplementary Information. Extra data are available from the corresponding authors on request.

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

## Acknowledgements

This project was financially supported by the National Natural Science Foundation of China (22173055 awarded to L.D., 22273057 awarded to M.D.L.), Innovation Team Project (2019KCXTD007 awarded to M.D.L.) of the Educational Commission of Guangdong Province of China, the Universities Joint Laboratory of Guangdong, Hong Kong and Macao (2021LSYS009 awarded to M.D.L.) and the Natural Science Foundation of Guangdong Province (2022A1515011661 awarded to M.D.L., 2023A1515012631 awarded to L.D.).

## Author contributions

G.X.Y. conceived the study, synthesized the materials, and performed RTP mechanism and application studies. L.D. supervised the project. G.X.Y., S. B.H., X.D., X.L.S., W.H., B.S., L.D., and M.D.L. discussed the results and edited the manuscript.

## Competing interests

The authors declare no competing interests.
