## [Peer Review File · Nature Communications]

Efficient intersystem crossing and tunable ultralong organic room-temperature phosphorescence via doping polyvinylpyrrolidone with polyaromatic hydrocarbonsREVIEWER COMMENTS

Reviewer #1 (Remarks to the Author):

The authors prepared various tunable ultralong RTP materials via doping polyaromatic hydrocarbons with polyvinylpyrrolidone. The authors conducted a detailed research and analysis. However, this work still has some flaws, and I think this work can be published on Nature Communications, only after major revisions as pointed out below.

1. Page 2 line 46, the first letter of Benzophenone should be lowercase. The expression of the entire text needs further verification.
2. Some typical literatures of polymer-based RTP materials with color tunable afterglow and near-infrared emission should be cited in the article to be more convincing, such as Adv. Mater. 2022, 34, 2108333; Chem. Eng. J. 2023, 469, 143931; Eur. Polym. J. 2024, 202, 112600.
3. Fig. 2b does not show a clear trend in afterglow variation. The afterglow videos of the doped PVP films should be added in the supporting information. 4. The quantum yield of the doped PVP films should be measured.

Reviewer #2 (Remarks to the Author):

Manuscript entitled “Efficient intersystem crossing and tunable ultralong organic room temperature phosphorescence via doping polyaromatic hydrocarbons with polyvinylpyrrolidone” reports on the preparation of ORTP materials with tunable phosphorescent colors and long lifetimes by the polymer doping strategy. Specifically, doping of polyvinylpyrrolidone with polyaromatic hydrocarbons results in effective ORTP materials to be used in data encryption, 3D printing and programmable labels. The work certainly falls in a current hot research topic and offers a simple strategy to prepare ORTP materials, however there are some points which need improvements (from very minor to major) before publication.

- 1) We would suggest to modify the title into: “Efficient intersystem crossing and tunable ultralong organic room temperature phosphorescence via doping polyvinylpyrrolidone with polyaromatic hydrocarbons”
- 2) The introduction paragraph needs to be better written and organized: some sentences need to be anticipated, others postponed. Moreover: “...due to **special** luminescent time.” The adjective special is meaningless...; “According to the Jablonski diagram, there are two **strategies** to achieve good phosphorescence: one is to promote the intersystem crossing (ISC) rate from excited singlet state to excited triplet state; the other is to enhance the stabilities of triplet exciton of **organic** molecules.” **strategies** should be changed into “requirements”, moreover these requirements hold for every phosphorescent material; “One is that lower T1 implies a larger band gap (ΔE_{st}) between S1 and T1, which is not favorable to intersystem crossing (ISC) of excitons, **not to say phosphorescence**. The other is that lower T1 is more likely to lead to non-radiative depletion of excitons, which results in a significant reduction in the lifetime and intensity of the phosphorescence.” What does “**not to say**

phosphorescence“ mean?; “This study gives us the hint that to make good use of efficient fluorescent molecules such as aromatic hydrocarbons as the guests, which are highly conjugated with very low T1 energy levels which is essential for long-wavelength phosphorescence.” Please re-write; why Sn has been added in the left part of Fig. 1a? this is misleading..

3) Results and discussion paragraph: “First of all, when different fluorescent PAHs molecules as guests (Fig. 2a) and PVP (Mw = 40,000 45,000 g/mol) as host are mixed at a mass ratio of 1% to prepare **polymer films**, and fluorescence emission under 365 nm UV excitation is observed (Fig. 2b UV on and Supplementary Fig. 1).” **Polymer films** should be better substituted by “blended films” or “doped films”; “Moreover, the luminescent lifetimes gradually increase with longer irradiation time (Fig. 2d). In the meantime, good to excellent phosphorescence quantum yields (Φ_p) of these activated films are determined and listed in Table 1.” What is the irradiation time at which values listed in table 1 are given?; sentence “BePh/PVP films and found that the RTP was further enhanced under vacuum conditions (Supplementary Fig. 4). The weak phosphorescence of the pristine samples may be caused by the energy transfer from the triplet excitons of the guest compound to the O2 molecule in the PVP. These results conversely indicate that the luminescence involves triplet **states** and explain the results in Fig. 2c and 2d that the luminescent intensities and lifetimes are enhanced along with the increase of irradiation time allowing O2 consumption.” Is not clear at all: “The weak phosphorescence of the pristine samples may be caused by the energy transfer from the triplet...” means “**The quenching** of the phosphorescence of the pristine samples may be caused by the energy transfer from the triplet...”?? the use of “conversely” is not clear; “1 vinyl 2 pyrrolidinone (VP) as a monomer for PVP has higher highest occupied molecular orbital (HOMO) and **lowest** unoccupied molecular orbital (LUMO) energy levels than PVP.”; regarding Fig. 2c (Delayed emission spectra of these doped PVP films after photoactivation under UV irradiation at 365 nm at 105 different times, delay time: 1 ms) due to impurity/reproducibility concerns it is reasonable to wonder if spectra have been collected for different samples; “); “The energy differences between the HOMOs of the guests and LUMO of PVP could be small enough for intermolecular charge transfer (Supplementary Fig. 11).” Which is a reasonable limit? A reference should be added; “To validate the above experimental data, the excited state calculations have been performed on the guests and the **complexes**.” Wouldn’t it be better to use the word “pair” or “dimer”?; “Therefore, the charge transfer state opens a new trajectory for the ISC of the guests, which will dramatically improve the ISC process of PAHs guest molecules. More importantly, the calculated triplet state energies essentially match the phosphorescence colors, which also **suggests** that the phosphorescence emission originates from the guest triplet state.” We would change suggests with confirms or supports since experiments in solution at 77 K have already supported this mechanism; “After 2 minutes of irradiation, the UV light is removed, and the drawn patterns relevant afterglows **respectively** (Fig. 5a and Supplementary Fig. 12).”; “By mask illumination of the samples, any pattern can be applied as phosphorescent information storage. Py/PVP and FAn/PVP films are covered with mask A and then irradiated by the 365 nm UV light (9 mWcm⁻²) for 120 seconds. **Remove the mask after the writing is completed, and print a Chinese character ‘ 鑫 ’ on the film.**”; “Afterward, the pattern is erased via annealing at **80 °C for 5 min**, and a renewed film is obtained by replacing mask A with mask B and repeating the above procedures, the quick response code can be clearly printed.” The annealing at 80°C for 5 min to erase the pattern is mentioned “suddenly” and for the first time at this point of the manuscript: how has this temperature been selected? Have thermal analysis of the films been performed?; “Among them, the RTP emissions of Py/PVP, BeAn/PVP, DBCh/PVP and BePe/PVP materials are all in the red region with RTP lifetimes 198 of 358, 139, 468 and 214 ms, and afterglow durations of **4, 3, 7 and 10 s**, respectively, which are very

promising promising red RTP emitters with exceptional performance.” These values seem “strange”.

4) Methods paragraph: “Unless otherwise stated, all materials are purchased from commercial sources without further **eradication**.”; “**Grinding powder** is used for all tests, and the amounts of samples for PL measurement are similar. Time resolved PL decay kinetics and PL spectra **were recorded in different temperatures by the Edinburgh company**.”

5) Supporting: Figure S4: annealed at which temperature? In the manuscript the annealing is performed to erase the label...; Fig S5 right “absorance”; “Table. S1. Energies of S1, T1, T2, Energy Gaps (in eV) between S1 and T1/T2 States and SOC (**Inter-system crossing rate constant**) of ...” these are spin orbit coupling costants....; SOC values of VP in Table 1 are “strange”; Fig. S13b: what is the difference with Fig. S4 in the annealing procedure????; Fig. S8: Py/PMMA or Py/KBr????

6) References 59 and 60 are lacking

Reviewer #3 (Remarks to the Author):

This manuscript by Li Dang et al. provided a polymer-based RTP system using PAHs as the dopants and the PVA as the matrix. Though doping system using PAHs such as Py and Co have been wildly reported, this work focuses on the effect of PVP polymer on dopants. The authors provided an improved mechanism using 1CT/3CT as a step between S_n and T₁ of the guest. They attempted to prove through transient absorption and theoretical calculations that the 1-vinyl-2-pyrrolidinone (VP) unit and guest formed the CT states. However, the evidence provided by the authors is insufficient, and I believe that non aromatic VP cannot form a CT state with a PAH. After abandoning the mechanism of polymer matrix providing CT effect, this work only introduced a polymer-based RTP system, which obviously lacks sufficient novelty. Therefore, I would like reject this manuscript.

1) The TA spectra only show guest radical cation. I think this evidence is not enough for the host-guest CT states, because CT absorption and power-law decay were not observed.

2) CT states consists one host and one guest molecule. But the theoretical calculation is based on a set of several VP molecule. I think it doesn't make sense. For one VP molecule, it is impossible to form a CT state with a PAH.

3) The authors says “...SOC values are quite small ... that ISC is difficult to occur...”. Actually, most of these guests can show RTP after doped into polymers such as PVA without the so-called CT states.

4) Even if the CT states exists, the proposed mechanism (figure 4d) is incorrectly expressed. It is “energy transfer” from 1CT/3CT to T₁, not “charge recombination”. And 1CT/3CT is generated through a serious steps containing charge separation, charge transfer and charge recombination.

5) Based on the doping principle of different guests and the decided host molecule (pyrrolidone), how to verify the best intermolecular distance and interactions between host and guest? Because the author only chose a mass ratio of 1% to prepare polymer films with host molecule of the same molecular weight.

Reviewer #4 (Remarks to the Author):

■ Detailed Responses to the Reviewer #1's Comments:

Comment 1: Page 2 line 46, the first letter of Benzophenone should be lowercase. The expression of the entire text needs further verification.

Response: We are sorry for this mistake, we have corrected the description (page 2 in the revised text) as “These materials can be easily fabricated by mixing pyrene (electron donor) and benzophenone (electron acceptor) using the melt-casting method.” In addition to this, we corrected all typos and polished the whole manuscript based on the reviewers' suggestions. We highlighted the corrections in the revised manuscript and thank the reviewers for this suggestion.

Comment 2: Some typical literatures of polymer-based RTP materials with color tunable afterglow and near-infrared emission should be cited in the article to be more convincing, such as *Adv. Mater.* 2022, 34, 2108333; *Chem. Eng. J.* 2023, 469, 143931; *Eur. Polym. J.* 2024, 202, 112600.

Response: Many thanks to the reviewer for your suggestion, which we feel is very reasonable. Therefore, after carefully reading the relevant literature suggested by the reviewer, we have cited them in the appropriate places. (page 3 in the revised text) as “More to the point, only sporadic examples have emitted ultra-long wavelength afterglows so far,⁵⁶⁻⁵⁸ and most of them are organic crystals.^{1,2,59-61}”

56.Kong, L. et al. Tunable ultralong multicolor and near-infrared emission from polyacrylic acid-based room temperature phosphorescence materials by FRET. *Chem. Eng. J.* **469**, 143931 (2023).

57.Lin, F. et al. Stepwise Energy Transfer: Near-Infrared Persistent Luminescence from Doped Polymeric Systems. *Adv. Mater.* **34**, 2108333 (2022).

58.Zhu, Y. et al. Achieving color-tunable persistent afterglow from ultralong polyacrylamide-based room-temperature phosphorescence materials through phosphorescence Förster resonance energy transfer. *Eur. Polym. J.* **202**, 112600 (2024).

Comment 3: Fig. 2b does not show a clear trend in afterglow variation. The afterglow videos of the doped PVP films should be added in the supporting information.

Response: We thank the reviewer for this reminder. We have added the afterglow

videos of the doped PVP films in the supporting information as suggested, as shown in Supplementary Video.

Comment 4: The quantum yield of the doped PVP films should be measured.

Response: We sincerely thank the review for the valuable suggestions, and the photophysical data Table 1 for doped PVP films has been added to the manuscript as suggested.

Table 1. Photophysical data of the polymer based ORTP materials.

Sample	Fluo.			Phos.		
	$\lambda_{em}(nm)$	$\phi_F(\%)$	$\tau(ns)$	$\lambda_{em}(nm)$	$\phi_P(\%)$	$\tau(ms)$
BecPh/PVP	422	14.3	40	550	27.5	1037
BePh/PVP	405	13.9	15	552	12.9	849
FIAn/PVP	467	22.2	41	585	7.7	255
Py/PVP	416	11.2	62	600	8.4	358
BeAn/PVP	413	7.1	36	606	6.2	139
Pi/PVP	425	7.6	28	553	10.3	766
BeTe/PVP	421	15.4	28	606	13.8	582
DBeCh/PVP	418	13.4	12	629	24.3	468
BePe/PVP	432	19.5	67	627	7.3	214
Co/PVP	447	11.1	86	566	10.1	3850

Ex. of Fluo.: 375 nm; Ex. of Phos.: 365 nm; Delayed time: 1 ms.

■ Detailed Responses to the Reviewer #2's Comments:

Comment 1: We would suggest to modify the title into: “Efficient intersystem crossing and tunable ultralong organic room temperature phosphorescence via doping polyvinylpyrrolidone with polyaromatic hydrocarbons”

Response: Thanks to the reviewer for your suggestion. We have changed the title of the manuscript to “Efficient intersystem crossing and tunable ultralong organic room temperature phosphorescence via doping polyvinylpyrrolidone with polyaromatic hydrocarbons”.

Comment 2: The introduction paragraph needs to be better written and organized: some sentences need to be anticipated, others postponed. Moreover: “...due to **special**

luminescent time.” The adjective special is meaningless...; “According to the Jablonski diagram, there are two **strategies** to achieve good phosphorescence: one is to promote the intersystem crossing (ISC) rate from excited singlet state to excited triplet state; the other is to enhance the stabilities of triplet exciton of **organic** molecules.” **strategies** should be changed into “requirements”, moreover these requirements hold for every phosphorescent material; “One is that lower T_1 implies a larger band gap (ΔE_{st}) between S_1 and T_1 , which is not favorable to intersystem crossing (ISC) of excitons, **not to say phosphorescence**. The other is that lower T_1 is more likely to lead to non-radiative depletion of excitons, which results in a significant reduction in the lifetime and intensity of the phosphorescence.” What does “**not to say phosphorescence**” mean? “This study gives us the hint that to make good use of efficient fluorescent molecules such as aromatic hydrocarbons as the guests, which are highly conjugated with very low T_1 energy levels which is essential for long-wavelength phosphorescence.” Please re-write; why S_n has been added in the left part of Fig. 1a? this is misleading.

Response: We sincerely thank the reviewer for careful reading.

(1) The word “special” has been deleted.

(2) The word “strategies” has been revised as “requirements”.

(3) Since phosphorescence originates from the radiative transition of excited triplet excitons. But when the band gap (ΔE_{st}) between S_1 and T_1 is too large, it is difficult for ISC to occur, and it is difficult to produce triplet excitons. This makes it difficult to observe phosphorescence. Therefore, we have revised “not to say phosphorescence” to “making it difficult to observe phosphorescence.” (page 2).

(4) We have revised “This study gives us the hint that to make good use of efficient fluorescent molecules such as aromatic hydrocarbons as the guests, which are highly conjugated with very low T_1 energy levels which is essential for long-wavelength phosphorescence.” to “This study gives us the hint that to make good use of polyaromatic hydrocarbons as the guests with very low T_1 energy levels due to the highly conjugated nature of these molecules, which is crucial for long-wavelength phosphorescence.” (page 2).

(5) Thanks to the reviewer for this reminder. We have corrected the Fig. 1a as follows.

Comment 3: Results and discussion paragraph: “First of all, when different fluorescent PAHs molecules as guests (Fig. 2a) and PVP (Mw = 40,000-45,000 g/mol) as host are mixed at a mass ratio of 1% to prepare polymer films, and fluorescence emission under 365 nm UV excitation is observed (Fig. 2b UV on and Supplementary Fig. 1).” polymer films should be better substituted by “blended films” or “doped films”; “Moreover, the luminescent lifetimes gradually increase with longer irradiation time (Fig. 2d). In the meantime, good to excellent phosphorescence quantum yields (Φ_p) of these activated films are determined and listed in Table 1.” What is the irradiation time at which values listed in table 1 are given; sentence “BePh/PVP films and found that the RTP was further enhanced under vacuum conditions (Supplementary Fig.4). The weak phosphorescence of the pristine samples may be caused by the energy transfer from the triplet excitons of the guest compound to the O₂ molecule in the PVP. These results conversely indicate that the luminescence involves triplet states and explain the results in Fig. 2c and 2d that the luminescent intensities and lifetimes are enhanced along with the increase of irradiation time allowing O₂ consumption.” Is not clear at all: “The weak phosphorescence of the pristine samples may be caused by the energy transfer from the triplet...” means “The quenching of the phosphorescence of the pristine samples may be caused by the energy transfer from the triplet..”? the use of “conversely” is not clear; “1-vinyl-2-pyrrolidinone (VP) as a monomer for PVP has higher highest occupied molecular orbital (HOMO) and lowest unoccupied molecular orbital (LUMO) energy levels than PVP.”; regarding Fig. 2c (Delayed emission spectra of these doped PVP

films after photoactivation under UV irradiation at 365 nm at 105 different times, delay time: 1 ms) due to impurity/reproducibility concerns it is reasonable to wonder if spectra have been collected for different samples; “The energy differences between the HOMOs of the guests and LUMO of PVP could be small enough for intermolecular charge transfer (Supplementary Fig. 11).” Which is a reasonable limit? A reference should be added; “To validate the above experimental data, the excited state calculations have been performed on the guests and the complexes.” Wouldn’t it be better to use the word “pair” or “dimer”? “Therefore, the charge transfer state opens a new trajectory for the ISC of the guests, which will dramatically improve the ISC process of PAHs guest molecules. More importantly, the calculated triplet state energies essentially match the phosphorescence colors, which also suggests that the phosphorescence emission originates from the guest triplet state.” We would change suggests with confirms or supports since experiments in solution at 77 K have already supported this mechanism; “After 2 minutes of irradiation, the UV light is removed, and the drawn patterns relevant afterglows respectively (Fig. 5a and Supplementary Fig. 12).”; “By mask illumination of the samples, any pattern can be applied as phosphorescent information storage. Py/PVP and FlAn/PVP films are covered with mask A and then irradiated by the 365 nm UV light (9 mWcm^{-2}) for 120 seconds. Remove the mask after the writing is completed, and print a Chinese character ‘鑫’ on the film.”; “Afterward, the pattern is erased via annealing at 80 for 5 min, and a renewed film is obtained by replacing mask A with mask B and repeating the above procedures, the quick response code can be clearly printed.” The annealing at 80°C for 5 min to erase the pattern is mentioned “suddenly” and for the first time at this point of the manuscript: how has this temperature been selected? Have thermal analysis of the films been performed? “Among them, the RTP emissions of Py/PVP, BeAn/PVP, DBCh/PVP and BePe/PVP materials are all in the red region with RTP lifetimes 198 of 358, 139, 468 and 214 ms, and afterglow durations of 4, 3, 7 and 10 s, respectively, which are very promising red RTP emitters with exceptional performance.” These values seem “strange”.

Response: Thanks to the reviewer's suggestions.

- (1) The phrase “polymer films” has been revised as “doped films”.
- (2) The phosphorescence quantum yields (Φ_p) of these doped films listed in Table 1 were determined after they were photoactivated for the specific radiation durations shown in Fig. 2c;
- (3) We have revised “The weak phosphorescence of the pristine samples may be caused by the energy transfer from the triplet excitons of the guest compound to the O₂ molecule in the PVP. These results conversely indicate that the luminescence involves triplet states and explain the results in Fig. 2c and 2d that the luminescent intensities and lifetimes are enhanced along with the increase of irradiation time allowing O₂ consumption.” to “The quenching of the phosphorescence of the pristine samples without photoactivation may be caused by the energy transfer from the triplet excitons of the guest compound to the O₂ molecule in the PVP. These results also indicate that the luminescence involves triplet states and explain the results in Fig. 2c and 2d that the luminescent intensities and lifetimes are enhanced along with the increase of irradiation time allowing O₂ consumption.” (page 3).
- (4) We have revised “1-vinyl-2-pyrrolidinone (VP) as a monomer for PVP has higher highest occupied molecular orbital (HOMO) and lowest unoccupied molecular orbital (LUMO) energy levels than PVP.” to “1-vinyl-2-pyrrolidinone (VP) as a monomer for PVP has higher highest occupied molecular orbital (HOMO) and lower lowest unoccupied molecular orbital (LUMO) energy levels than PVP.” (page 6).
- (5) Thanks to the reviewer's kind reminder, we re-prepared the doped films after checking the purity of the reagent and tested the delayed spectra and still obtained the same results as in Fig. 2c.
- (6) Thanks to the reviewers for this question, we have fully considered the effect of the energy difference between the HOMOs of the guest and the LUMO of the PVP on its ability to undergo intermolecular charge transfer. When the energy difference between the HOMO of the guest and the LUMO of the host is too small, it causes intermolecular charge transfer in the ground state which is observed in some organic cocrystal photothermal materials, it is more favorable to undergo the non-irradiation process rather than luminescence. Only when the energy difference between the HOMO of the

guest and the LUMO of the host is appropriate, the intermolecular charge transfer occurs in the excited state after being excited, which is more favorable for its luminescence. We have corrected the description (page 6 in the revised text) as “The energy differences between the HOMOs of the guests and LUMO of PVP could be small enough for intermolecular charge transfer^{64,65}.”

64. Gould, I.R., Young, R.H., Mueller, L.J., Albrecht, A.C. & Farid, S. Electronic Structures of Exciplexes and Excited Charge-Transfer Complexes. *J. Am. Chem. Soc.* **116**, 8188-8199 (1994).

65. Jenekhe, S.A. & Osaheni, J.A. Excimers and Exciplexes of Conjugated Polymers. *Sinence* **265**, 765-768 (1994).

(7) We have revised “To validate the above experimental data, the excited state calculations have been performed on the guests and the complexes.” to “To validate the above experimental data, the excited state calculations have been performed on the guests and the pair.” (page 6).

(8) We have revised “More importantly, the calculated triplet state energies essentially match the phosphorescence colors, which also suggests that the phosphorescence emission originates from the guest triplet state.” to “More importantly, the calculated triplet state energies essentially match the phosphorescence colors, which also confirms that the phosphorescence emission originates from the guest triplet state.” (page 7).

(9) We have revised “After 2 minutes of irradiation, the UV light is removed, and the drawn patterns relevant afterglows respectively (Fig. 5a and Supplementary Fig. 12).” to “After 2 minutes of irradiation, the UV light is removed, and the patterns are drawn with corresponding afterglows.” (page 7).

(10) We have revised “Py/PVP and FIAN/PVP films are covered with mask A and then irradiated by the 365 nm UV light (9 mWcm^{-2}) for 120 seconds. Remove the mask after the writing is completed, and print a Chinese character ‘鑫’ on the film.” to “Py/PVP and FIAN/PVP films are covered with mask A of the Chinese character ‘鑫’ and then irradiated by the 365 nm UV light (9 mWcm^{-2}) for 120 seconds. Remove the mask after the writing is completed, and the corresponding afterglow of the Chinese character ‘鑫’ appears on the doped film when the information is read.” (page 8).

(11) We sincerely thank the reviewer for this kind reminder. We performed the thermal

analysis of the doped films, and four annealing temperatures, 20 °C, 40 °C, 60 °C and 80 °C were selected. Different annealing temperatures required different annealing times. When the annealing temperature increases, the required annealing time decreases, which satisfies the principle of time-temperature equivalence. In order to achieve faster annealing results, we finally chose an annealing temperature of 80 °C and an annealing time of 5 minutes.

The following sentences “In addition, four annealing temperatures, 20 °C, 40 °C, 60 °C and 80 °C were selected for thermal analysis of the doped films. As the annealing temperature increases, the required annealing time decreases, which is in accordance with the time-temperature equivalence principle. In order to achieve faster annealing results, an annealing temperature of 80 °C and an annealing time of 5 minutes were chosen.” have been inserted in the revised manuscript on page 3.

(12) We are sorry for this mistake. we have corrected the description (page 8) as “Among them, the RTP emissions of Py/PVP, BeAn/PVP, DBCh/PVP and BePe/PVP materials are all in the red region with RTP lifetimes 358, 139, 468 and 214 ms and afterglow durations of 4, 3, 7 and 4 s, respectively, which are very promising red RTP emitters with exceptional performance.

Comment 4: Methods paragraph: “Unless otherwise stated, all materials are purchased from commercial sources without further eradication.”; “Grinding powder is used for all tests, and the amounts of samples for PL measurement are similar. Time-resolved PL decay kinetics and PL spectra were recorded in different temperatures by the Edinburgh company.”

Response: We sincerely thank the reviewer for careful reading.

(1) We have revised “Unless otherwise stated, all materials are purchased from commercial sources without further eradication.” to “All other solvents and reagents were purchased with analytical grade and used without further purification.” (page 9).

(2) We have revised “Grinding powder is used for all tests, and the amounts of samples for PL measurement are similar. Time-resolved PL decay kinetics and PL spectra were

recorded in different temperatures by the Edinburgh company.” to “Doped films are used for all tests, and the amounts of samples for PL measurement are similar. Time-resolved PL decay kinetics and PL spectra were recorded in room temperature by the FLS980 from Edinburgh company”. (page 9).

Comment 5: Supporting: Figure S4: annealed at which temperature? In the manuscript the annealing is performed to erase the label...; Fig S5 right “absorance”; “Table. S1. Energies of S₁, T₁, T₂, Energy Gaps (in eV) between S₁ and T₁/T₂ States and SOC (Inter-system crossing rate constant) of ...” these are spin-orbit coupling constants....; SOC values of VP in Table 1 are “strange”; Fig. S13b: what is the difference with Fig. S4 in the annealing procedure?; Fig. S8: Py/PMMA or Py/KBr?

Response: Thank you to the reviewer for the suggestions.

(1) The annealing temperature is 80 °C.

(2) We have corrected the Fig. S5 as suggested.

(3) We are very sorry for this mistake. We have revised “Intersystem crossing rate constant” to “spin-orbit coupling constants”. In addition, we checked the calculation of the SOC value of the VP which is the same as in the table.

(4) Fig. S4 shows the effect of photoactivation and annealing treatment on the luminescence properties of the doped films under vacuum conditions and in air, where only one annealing treatment is performed. Fig. S16b explores the reproducibility of the activation-deactivation process of the doped film, so multiple photoactivation and annealing treatments are performed (annealing temperature: 80 °C, annealing time:

5 minutes.)

(5) We are very sorry for this mistake. We have tested both Py/PMMA and Py/KBr samples in our experiments, but we have only discussed the Py/PMMA's results in the manuscript.

Comment 6: References 59 and 60 are lacking.

Response: Thank you to the reviewer for the suggestions. We have added these two references in the manuscript.

62. Badger, B. & Brocklehurst, B. Absorption spectra of dimer cations. Part 3.—Naphthalene and anthracene derivatives and pyrene. *Trans. Faraday Soc.* 65, 2588-2594 (1969).

63. Mori, Y., Shinoda, H., Nakano, T. & Kitagawa, T. Formation and Decay Behaviors of Laser-Induced Transient Species from Pyrene Derivatives 1. Spectral Discrimination and Decay Mechanisms in Aqueous Solution. *J. Phys. Chem. A* 106, 11743-11749 (2002).

■ Detailed Responses to the Reviewer #3's Comments:

Comment 1: The TA spectra only show guest radical cation. I think this evidence is not enough for the host-guest CT states, because CT absorption and power-law decay were not observed.

Response: We sincerely thank the reviewer for this question.

For the CT between host and guest in Py/PVP doped films, the population and transformation of the CT absorption is usually very fast. As shown in Fig.S7, the transient absorption peaks gradually rise up and have a red-shift from 433 to 441 nm for Py/PVP film and from 521 to 525 nm for BePe/PVP film at the beginning delay times. These steps are attributed to the CT process in the doped film.

In doped systems, where the host is the vast majority and the guest content is very small, it is usually very difficult to observe the cationic or anionic radicals by the transient absorption when it undergoes intermolecular charge transfer in the excited state. But in our system, the conjugation of the guest molecule is very good and aromatic, so we can observe the cationic radical of the guest. However, the host molecule is not conjugated,

so the molar extinction coefficient of its anionic radicals will be relatively small, it is difficult to detect the corresponding anionic cations. Nevertheless, the detection of the cationic radical signal of the guest molecule in the transient absorption spectra can further prove that the intermolecular charge transfer process in the excited state has occurred between host and guest in the doped systems.

Fig. S7. Shown are fs-TA spectra of Py/PVP (a) and BePe/PVP (b) films at the beginning delay times after the activation with 365 nm UV light, respectively.

Revision: To address this question, the following sentences have been inserted into the manuscript on page 4.

For the CT between host and guest in Py/PVP doped films, the population and transformation of the CT absorption is usually very fast. As shown in Supplementary Fig.7, the transient absorption peaks gradually rise up and have a red-shift from 433 to 441 nm for Py/PVP film and from 521 to 525 nm for BePe/PVP film at the beginning delay times. These steps are attributed to the CT process in the doped film. Generally speaking, the host is the vast majority and the guest content is very small in the doped systems. Therefore, it is usually very difficult to observe the cationic or anionic radicals by the transient absorption when it undergoes intermolecular charge transfer in the excited state. But in our system, the conjugation of guest molecules is very good and aromatic, thus the cationic radical of the guest is directly observed by fs-TA. However, the host molecule is not conjugated, thus the molar extinction coefficient of its anionic radicals will be relatively small, it is difficult to detect the corresponding anionic cations.

Nevertheless, the detection of the cationic radical signal of the guest molecule in the transient absorption spectra can further prove that the intermolecular charge transfer process in the excited state has occurred between host and guest in the doped systems.

Comment 2: CT states consists one host and one guest molecule. But the theoretical calculation is based on a set of several VP molecule. I think it doesn't make sense. For one VP molecule, it is impossible to form a CT state with a PAH.

Response: We sincerely thank the reviewer for this question. For the small molecular doping system, the CT state should have consisted of a host molecule and a guest molecule. However, for this doping system, the host is a polymer. Since it is still very difficult to calculate and analyze the molecular orbital energy level of a polymer, the main purpose of our theoretical calculation is to reveal the trend of its molecular orbital energy level. With the increase of monomer polymerization, the HOMO energy level of the polymer gradually increases and the LUMO energy level gradually decreases, which increases the probability of intermolecular charge transfer with the guest PAHs in the excited state.

In addition, we strongly agree with the comment made by the reviewer that it is impossible for a VP molecule to form the CT state with a PAH according to our calculations. However, in the actual doping system, it is not the single VP molecule but the side groups of the PVP that have to form a CT state with the guest PAHs.

Revision: To address this question, the following sentences have been inserted into the manuscript on page 6.

For this doping system, the host is a polymer. Since it is very difficult to calculate and analyze the molecular orbital energy level of polymer, we select a monomer (1-vinyl-2-pyrrolidinone (VP)) of PVP as the template for analysis. The results show that VP...

Comment 3: The authors say "...SOC values are quite small ... that ISC is difficult to occur...". Actually, most of these guests can show RTP after doped into polymers such as PVA without the so-called CT states.

Response: Thanks to the reviewer for this question. This question is not only very

important, but also the key to distinguish our study from other similar studies. From the theoretical calculations, not only the SOC values of the PAHs are very small, close to 0, but also the ΔE_{st} is very large, which implies that it is extremely difficult for ISC to occur, but when the fs-TA spectra of such molecules were tested, anomalies were found. Taking the fs-TA spectra of Py and BePe as an example, we found the characteristic absorption peak of the excited triplet state of Py at 411 nm for Py in MeCN; we found the characteristic absorption peak of the excited triplet state of BePe at 470 nm for BePe in DCM. In addition, similar results were obtained for other PAHs molecules when fs-TA experiments were performed. This implies that PAHs molecules can actually undergo ISC to produce the corresponding excited triplet states, which is completely inconsistent with the calculated results. Through literature review⁶⁶⁻⁷⁰ and further experiments, we have revealed the reason why PAHs can undergo ISC. For PAHs molecules, whose structures are all planar, they are very prone to generate excimers upon excitation, which allows them to undergo ISC, leading to the generation of the corresponding triplet excitons. To test the hypothesis, we have measured the fs-TA spectra of Py with different concentrations in MeCN. The results found that when the concentration of Py is very small, the transient absorption signal of excited triplet state excitons of Py produced is very small because it is very difficult to form the excimers. However, with the increasing concentration of Py, it is more likely to produce excimers, so the intensity of the characteristic absorption peak of Py's excimer (508 nm) obviously increases, and the characteristic absorption peak of excited triplet states (411 nm) of Py is also obviously enhanced (Fig. S13a). In addition, we also tested the fs-TA spectra of BePe with different concentrations in DCM, and the same results can be obtained (Fig. S13b). But even so, the formation of the excimers based on the conjugated molecule promotes the ISC process only to a small extent, which explains the fact that most of these guests show very weak RTP when doped with polymers such as PVA, PMMA, etc. For example, the doped film of Py/PMMA produces a very weak red phosphorescence.

However, when excited state intermolecular charge transfer occurs between the host and the PAHs, the ISC process of PAHs is greatly facilitated, resulting in the generation

of a large number of triplet excitons. This can be verified not only from the small-molecule doping systems that we have previously studied (*J. Phys. Chem. Lett.* **2023**, *14*, 6927) but more importantly, the triplet state yield of the guest PAHs in the polymer system can also be dramatically boosted by its intermolecular charge transfer with the side groups of the PVP. This makes the RTP of Py/PVP doped films much stronger than that of Py/PMMA doped films at the same doping ratio (Fig. S14).

Fig.S13 (a) The fs-TA spectra of Py with different concentrations in MeCN after 320 nm excitation. (b) The fs-TA spectra of BePe with different concentrations in DCM after 365 nm excitation.

Fig.S14 Phosphorescence images of PMMA and PVP films doped with the same mass ratio of Py under UV light at 365 nm.

Ref:

66. Rodgers, M.A.J. Formation kinetics of the pyrene dimer cation observed by pulse radiolysis. *Chem. Phys. Lett.* **9**, 107-108 (1971).
67. Kira, A., Arai, S. & Imamura, M. Pyrene Dimer Cation as Studied by Pulse Radiolysis. *J. Chem. Phys.* **54**, 4890-4895 (1971).
68. Kryachko, E.S. Dicationic states of benzene dimer: Benzene dimer cation and benzene

dication parenthood patterns. *Int. J. Quantum Chem.* **107**, 2741-2755 (2007).

69. Getoff, N., Solar, S., Richter, U.-B. & Haenel, M.W. Pulse radiolysis of pyrene in aprotic polar organic solvents: simultaneous formation of pyrene radical cations and radical anions. *Radiat. Phys. Chem.* **66**, 207-214 (2003).

70. Kira, A., Nakamura, T. & Imamura, M. Pyrene—naphthalene complex radical cations. *Chem. Phys. Lett.* **54**, 582-584 (1978).

Revision: To address this question, the following sentences have been inserted into the manuscript on page 7.

But when the fs-TA spectra of such molecules were tested, anomalies were found. Taking the fs-TA spectra of Py and BePe as an example, the characteristic absorption peak of the excited triplet state of Py at 411 nm in MeCN and the excited triplet state of BePe at 470 nm for BePe in DCM can be detected. In addition, similar results were obtained for other PAHs molecules when fs-TA experiments were performed. This implies that PAHs molecules can actually undergo ISC to produce the corresponding excited triplet states, which is completely inconsistent with the calculated results. Through literature review⁶⁶⁻⁷⁰ and further experiments, we have revealed the reason why PAHs can undergo ISC. For PAHs molecules, whose structures are all planar, they are very prone to generate the excimers upon excitation, which allows them to undergo ISC, leading to the generation of the corresponding triplet excitons. To test the hypothesis, we have measured the fs-TA spectra of Py with different concentrations in MeCN. The results found that when the concentration of Py is very small, the transient absorption signal of excited triplet state excitons of Py is very small because it is very difficult to form the excimers. However, with the increasing concentration of Py, it is more likely to produce excimers, so the intensity of the characteristic absorption peak of Py's excimer (508 nm) obviously increases, and the characteristic absorption peak of excited triplet states (411 nm) of Py is also obviously enhanced (Supplementary Fig. 12a). In addition, we also tested the fs-TA spectra of BePe with different concentrations in DCM, and the same results can be obtained (Supplementary Fig. 12b). But even so, the formation of the excimers based on the conjugated molecule promotes the ISC process only to a small extent, which explains the fact that most of these guests show very weak RTP when doped with polymers such as PVA, PMMA, etc. For example, the

doped film of Py/PMMA produces a very weak red phosphorescence. However, when excited state intermolecular charge transfer is formed between the host and PAHs molecules, the ISC process of PAHs molecules is greatly facilitated, resulting in the generation of a large number of triplet excitons. This can be verified not only from the small-molecule doping systems that we have previously studied but more importantly, the triplet state yield of the guest PAHs in the polymer system can also be dramatically boosted by its intermolecular charge transfer with the side groups of the PVP. This makes the RTP of Py/PVP doped films much stronger than that of Py/PMMA doped films at the same doping ratio (Supplementary Fig. 13).

Comment 4: Even if the CT states exists, the proposed mechanism (figure 4d) is incorrectly expressed. It is “energy transfer” from $^1\text{CT}/^3\text{CT}$ to T_1 , not “charge recombination”. And $^1\text{CT}/^3\text{CT}$ is generated through a serious steps containing charge separation, charge transfer and charge recombination.

Response: Thank you for the reviewer's reminder and we fully agree with your comments. Charge transfer is the initiating force, and if there is no charge transfer, there is no $^1\text{CT}/^3\text{CT}$, but the conversion from $^1\text{CT}/^3\text{CT}$ to T_1 should have occurred with both charge recombination and energy transfer. Therefore, we have corrected Fig. 4d as you suggested.

Comment 5: Based on the doping principle of different guests and the decided host molecule (pyrrolidone), how to verify the best intermolecular distance and interactions

between host and guest? Because the author only chose a mass ratio of 1% to prepare polymer films with host molecule of the same molecular weight.

Response: We sincerely thank the reviewer for this suggestion. We verified the optimal intermolecular distance and interactions between host and guest by testing the phosphorescence efficiency. In our experiments, we prepared doped films with different mass ratios (guest-host mass ratios of 1/10, 1/100, 1/500, and 1/1000) for phosphorescence lifetime (τ_p) and phosphorescence quantum yield (Φ_p) testing. For example, the τ_p and Φ_p of Py/PVP films with mass ratios of 1/10, 1/100, 1/500 and 1/1000 were 347 ms 8.2%, 358 ms 8.4%, 332 ms 7.3%, and 325 ms 6.9%, respectively. Similar results were obtained for doped films with different PAHs as guests. Since 1% mass ratio of doped films gave the best phosphorescence results, we finally chose 1% mass ratio for the preparation of doped PVP films.

REVIEWERS' COMMENTS

Reviewer #1 (Remarks to the Author):

The author has modified it according to my comments, and it can be accepted.

Reviewer #2 (Remarks to the Author):

The authors have addressed all the points raised by the Reviewers and the quality of the manuscript is now sufficient to justify its publication

Reviewer #3 (Remarks to the Author):

The revised version can be accepted.

Reviewer #4 (Remarks to the Author):
